# Unusual Patterns of Lateral Scutes in Two Olive Ridley Turtles and Their Genetic Assignment to the Thai Andaman Sea Populations of *Lepidochelys olivacea* Eschscholtz, 1829

**DOI:** 10.3390/biology13070500

**Published:** 2024-07-04

**Authors:** Patcharaporn Kaewmong, Kongkiat Kittiwattanawong, Korakot Nganvongpanit, Promporn Piboon

**Affiliations:** 1Phuket Marine Biological Center, Phuket 83000, Thailand; marineanimal.vet@gmail.com (P.K.); kkongkiat@gmail.com (K.K.); 2The School of Veterinary Medicine, Faculty of Veterinary Medicine, Chiang Mai University, Chiang Mai 50100, Thailand; korakot.n@cmu.ac.th

**Keywords:** sea turtle, marine animal, carapace, morphology, mtDNA, haplotype, RAG2

## Abstract

**Simple Summary:**

In this study, we used both morphology and molecular genetic tools, namely mitochondrial DNA (mtDNA) control region and nuclear DNA (nDNA) exon involving recombination activating gene 2 (RAG2), to confirm the species of two sea turtles found in the Thai Andaman Sea that possessed unusual lateral scute patterns from the original species. Both turtles were identified as the olive ridley turtle (*Lepidochelys olivacea*). This finding has raised awareness about the need for exploration of rare phenotypes of carapace scute patterns in the olive ridley turtle populations that reside in Thai Seas and the Indo-Pacific region.

**Abstract:**

Two stranded *Lepidochelys*-like sea turtles were rescued from the Thai Andaman Sea coastline by veterinarians of the Phuket Marine Biological Center (PMBC), one in May of 2019 and another in July of 2021. They were first identified as olive ridley turtles (*Lepidochelys olivacea*), as the external appearance of both turtles was closer to that species than the other four species found in the Thai Andaman Sea. In fact, when carefully examined, an unusual pattern of the lateral scutes on each turtle was observed, specifically symmetric 5/5 and asymmetric 5/6, both of which are considered rare for *L. olivacea* and had never been reported in the Thai Andaman Sea. In contrast, this characteristic was more common for the closely related species, Kemp’s ridley (*L. kempii*), although this species is not distributed in the Indo-Pacific Ocean. Thus, we further investigated their genetic information to confirm species identification using two molecular markers, namely the mtDNA control region and nDNA RAG2. The results from the mtDNA control region sequences using the Basic Local Alignment Search Tool (BLAST) indicated that both individuals exhibited a higher percent identity with *L. olivacea* (99.81–100.00%) rather than *L. kempii* (94.29–95.41%) or any other species. A phylogenetic tree confirmed that these two turtles belonged to the *L. olivacea* clade. Moreover, the results of RAG2 also supported the mtDNA result, as both individuals shared the same RAG2 haplotype with *L. olivacea*. Thus, we have concluded that the two turtles with unusual lateral scute patterns exhibited genetic consistency with their original species, *L. olivacea*, which has brought attention to the importance of exploring rare phenotypes in sea turtle populations residing in Thai Seas.

## 1. Introduction

Sea turtles are a migratory species that travel over large distances throughout the oceans of many countries [1,2]. Among the seven living species of sea turtles existing in the world, five species from two families are present in the seas of Thailand. These include the loggerhead (*Caretta caretta*), the hawksbill (*Eretmochelys imbricata*), the olive ridley (*Lepidochelys olivacea*), and the green turtle (*Chelonia mydas*), all from the family Cheloniidae, as well as the leatherback (*Dermochelys coriacea*) from the family Dermochelyidae [3,4]. All of these have been listed as protected marine animal species in Thailand [5]. Two other species, the Flatback (*Natator depressus*) and Kemp’s ridley turtle (*L. kempii*), have never been reported in this area [1,6].

Each sea turtle species has its own unique external morphological characteristics that include body shape, head shape, the number of frontal scales, the carapace form, the pattern of the scutes, and color. All of these features are commonly used to distinguish one species from another [1,7]. However, complications can sometimes arise for several reasons. For example, the unclear external appearance present on stranded and deceased turtles has often resulted from a high degree of decomposition or as a consequence of severe wounding [8,9]. A mixing of external characteristics due to hybridization can also be the cause of misidentification as they could present phenotypes of two species in one individual, such as the hybridization between *E. imbricata* × *C. caretta* and *E. imbricata* × *L. olivacea*, which has occurred in the Bahia State of Brazil [10,11]. In addition, Almeida, et al. [9] found that they could have external morphology identified as one species, *E. imbricata*, but had mitochondrial DNA (mtDNA) haplotypes as another, *C. caretta*. Moreover, the similarity in external morphology between related species could occur in some genera, such as the *Lepidochelys* species, *L. olivacea*, and *L. kempii*, which look similar to each other and can be confused when only morphology is used for identification [12].

*L. olivacea* has a relatively short and wide carapace with body coloring on the dorsal carapace that has been characterized as light to gray green or olive green, which clearly differentiates it from almost all other sea turtle species [1,13]. However, these features are also similar to those of a closely related species, *L. kempii*. Importantly, *L. olivacea* usually presents at least six to eight pairs of lateral scutes (costal scutes) [1], while five pairs are considered rare for this species. Only 79 individuals from a total of 655 newborn turtles observed in the natural nesting sites located along the southern coast of Sri Lanka have been recorded as having five symmetrical pairs of lateral scutes [14]. In fact, a pattern of five pairs of lateral scutes is more common for *L. kempii*. Another point to consider is that *L. olivacea* is predominantly distributed in the tropical waters of the Pacific, Indian, and Atlantic Oceans [15], while *L. kempii* is restricted to the Gulf of Mexico, along the Atlantic coast of the United States, and in the sea waters of western Europe [16]. Thus, the lateral scute pattern should not be used as the only criteria for identification of the *Lepidochelys* species [6,17]. This is particularly true in the overlapping areas for these two species in the Atlantic Ocean, which could easily lead to instances of misidentification. Because of certain complications, species identification based solely on body features may not always be entirely accurate. Thus, other useful tools, such as genetic data, could be used for sea turtle species identification, as well as external morphology involving mitochondrial DNA control regions and nuclear DNA markers [18,19,20]. 

In the Thai Andaman Sea, *L. olivacea* is one of three sea turtle species that can be found at this site [21,22]. It is the second-most abundant species in the area after *C. mydas* [21]. *L. olivacea* has been evaluated as a vulnerable species by the IUCN red list for global populations [15]; however, it is considered an endangered species in Thailand due to declines in their nesting numbers over the past decades. Their numbers have declined from 238 nests in 1979 to 42 nests in 1990 [13,22,23]. Recently, only five nests were reported in Phang Nga and Krabi Provinces during the period between 2009 and 2010 [21]. In addition, according to the stranding records of sea turtles in Thai seas from 2006 to 2015, a total of 210 olive ridley turtles were stranded, accounting for 19.72% of all stranded turtles in Thai sea waters [8]. Although the nesting areas and the stranding records along the Thai Andaman Sea coastline have been well documented [3,13,24], studies focusing on the external morphology, as well as records of carapace scute variations and genetic data for this species throughout this area, are considered rare. In accordance with Chantrapornsyl [13], common patterns of lateral scutes of *L. olivacea* found at Phra Thong Island, Thai Mueang, and Niyang Beach have been documented at mostly six to seven pairs, and sometimes up to nine pairs. Notably, asymmetrical patterns have also been reported in these areas [13]. 

Until now, the occurrence of *L. olivacea* with lateral scutes numbering less than six pairs had never been recorded in Thai seas, either in the Thai Andaman Sea or the Gulf of Thailand. Thus, in this study, we have provided the first record of the unusual lateral scute patterns of two individuals identified as *L. olivacea* that were found in the Thai Andaman Sea. The genetic data of both were revealed using the mtDNA control region (D-loop) and nuclear exon, along with recombination activating gene 2 (RAG2), to confirm species identification.

## 2. Materials and Methods

### 2.1. Sample and Morphological Data Collection

Two turtles in this study were rescued from the Thai Andaman Sea along the coast of Phuket Province at different times. The first individual (individual 1/sample GD) was stranded on 8 May 2019, while the second individual (individual 2/sample HT) was stranded on 6 July 2021. Since then, they have been cared for at the Phuket Marine Biological Center, Phuket, Thailand (PMBC). During an initial visual examination at the stranding site, both were identified as *L. olivacea* since they shared a similar external appearance to that species; however, unusual patterns of the lateral scute were observed on the carapace, which was considered rare for *L. olivacea* in this area. The effective identification of turtle species has relied on certain morphological characteristics, as outlined in the established guidelines [1,7]. Furthermore, dorsal and ventral photos were taken, while measurements of curved carapace length (CCL) and body weight were also recorded. 

### 2.2. DNA Extraction, Mitochondrial DNA Control Region, and Phylogenetic Tree

Blood samples of two individuals used in this study were left from the routine health checking program (leftover specimens). DNA extraction was performed using DNA extraction kits according to the manufacturer’s instructions (DNeasy^®^ Blood and Tissue Kit, QIAGEN, Hilden, Germany) at the Laboratory of the Faculty of Veterinary Medicine, Chiang Mai University. The mtDNA control region was amplified using one pair of primers: LCM15382 (5′-GCTTAACCCTAAAGCATTGG-3′) and H950g (5′-GTCTCGGATTTAGGGGTTTG-3′) [25]. PCR reactions were conducted in a total volume of 25 µL using a PCR thermocycler machine (Bio-Rad Laboratories, Inc., Hercules, CA, USA). The reactions consisted of 2.5 µL of 10× PCR buffer (160 mM (NH_4_)_2_SO_4_, 500 mM Tris–HCl, 17.5 mM MgCl_2_, and 0.1% Triton^TM^ X-100, Vivantis, Shah Alam, Selangor Darul Ehsan, Malaysia), 0.2 µL of *Taq* DNA polymerase (5 U/µL) (Vivantis, Selangor Darul Ehsan, Malaysia), 0.5 µL of 10 mM dNTP polymerase (Vivantis, Shah Alam, Selangor Darul Ehsan, Malaysia), 0.5 µL of 10 µM of each primer, and 0.5 µL of DNA templates (50 ng/µL). They were then adjusted to 25 µL with deionized water. The PCR conditions were performed as follows: 95 °C for 5 min; 40 cycles of denaturation at 95 °C for 30 s; an annealing step at 55 °C for 30 s; and extension at 72 °C for 1 min; 72 °C for 10 min. The PCR products were then sent for purification and sequencing analysis, which was performed by Macrogen (Seoul, Republic of Korea). Sequence identities were investigated for species identification using the Basic Local Alignment Search Tool (BLAST) that was made available at the National Center for Biotechnology Information (NCBI) GenBank. Sequences of the control region obtained in this study were deposited in GenBank (accession numbers PP908976 and PP908977). 

A phylogenetic tree of the mtDNA control region sequences was then constructed to observe the relationship of our samples among taxa using Bayesian analysis implemented in MrBayes version 3.2.7 [26]. Other sequences of *L. olivacea* and *L. kempii* were retrieved from the NCBI database (see all accession numbers used in Appendix A) [27,28,29,30,31,32,33,34]. The loggerhead sea turtle (*Caretta caretta*), accession number OR775090, was used as an outgroup. Accordingly, jModelTest version 2.1.10 was used to generate the best evolutionary tree model, which has been defined as GTR [35]. The phylogenetic tree was constructed on the run length of Markov Chain Monte Carlo (MCMC) at 4,000,000 iterations, using the average standard deviation of split frequencies below 0.01 as the convergence diagnostic. The robustness of each branch was assessed by the posterior probabilities (PP). The phylogenetic tree was then visualized using iTOL version 6.1.1 [36].

### 2.3. Variable Site Analysis of RAG2

Primer RAG2F (5′-CTGCTATCTTCCCCCTCTCC-3′) and RAG2R (5′-GTTGTCACACTGGTAGCCCC-3′) were used to amplify RAG2 [18]. PCR reactions were conducted in a total volume of 25 µL that contained PCR reagents, as has been described, for the purpose of control region amplification. The PCR conditions were performed as follows: 95 °C for 5 min; 40 cycles of denaturation at 95 °C for 30 s; an annealing step at 68 °C for 30 s; and extension at 72 °C for 1 min; 72 °C for 10 min. The PCR products were then sequenced by Macrogen (Seoul, Republic of Korea). RAG2 sequences obtained from our samples and other sequences of both *L. olivacea* and *L. kempii* from the previous study by Naro-Maciel, et al. [19] were included for the purpose of sequence alignment performed in MEGA-X version 10.2.2 [37]. The variable site was observed and determined using the DnaSP program version 6.12.3 [38]. Sequences of RAG2 obtained from this study were deposited in GenBank (accessions numbers PP916054 and PP916055). 

## 3. Results

### 3.1. Morphological Characteristics of Individual Turtles

Individual 1 measured 55.0 cm CCL and weighed 24.0 kg. Two pairs of prefrontal scales were observed (Figure 1A). Five pairs of lateral scutes (5/5) and five vertebral scutes were counted for this individual (Figure 1B). Four pairs of inframarginal scutes were noted for the plastron (Figure 1C). Individual 2 measured 18.5 cm CCL and weighed 0.6 kg. This individual also had two pairs of prefrontal scutes (Figure 2A), but the number of lateral scutes was asymmetric (Figure 2B). Five lateral scutes were observed on the left side of the carapace, though six were present on the right side (5/6). Six vertebral scutes were counted on the carapace. The plastron had four pairs of inframarginal scutes (Figure 2C). These two turtles clearly possessed some different external characteristics from the common *L. olivacea* in the Thai Andaman Sea, particularly with regard to their five lateral scutes. Thus, they were suspected to be *L. kempii*, or a hybrid species, that was not distributed in this area.

### 3.2. Mitochondrial DNA Control Region Sequence and Phylogenetic Tree

The mtDNA control region sequences of both individuals were consistent with *L. olivacea* found in the Indo-Pacific Ocean (i.e., GenBank accession numbers MN342242 and JN391463) at high percent identity within a range of 99.81 to 100.00% using BLAST, while the highest percent identity to *L. kempii* was limited at 94.29 to 95.41% (Appendix A). For phylogenetic analysis, the total length of the alignment mtDNA control region sequences used was 457 base pairs. It was determined that there were clear separate clades between *L. olivacea* and *L. kempii*, while our two samples were grouped within the clade *L. olivacea* (Figure 3). 

### 3.3. Sequences of RAG2

The total length of the RAG2 alignment dataset used in this study was 547 base pairs (see accession numbers in Table 1). There were two different haplotypes found between *L. olivacea* and *L. kempii*, with only one variable site found among this consensus alignment (Table 1). At position 180, a transitional site was found between *L. olivacea* (C) and *L. kempii* (T), although no differences in the nucleotides were observed among *L. olivacea* inhabiting the Pacific and Atlantic Oceans. Our samples also showed the same nucleotide as *L. olivacea* but not *L. kempii.*

## 4. Discussion

Based on morphological characteristics, the two individuals were suspected to be *L. kempii*, or a hybrid species comprised of *L. olivacea* × *L. kempii*, because of the presence of five paired lateral scutes and an asymmetric 5/6 lateral scute pattern. This feature is considered rare for *L. olivacea* inhabiting the Thai Andaman Sea, but it does resemble the common characteristics of *L. kempii*. However, *L. kempii* is not known to be distributed in other oceans apart from the Atlantic Ocean. Here, the other two molecular markers, the mtDNA control region and nuclear DNA exon RAG2, were successfully used to confirm the species. We have therefore concluded that these two stranded sea turtles in the Thai Andaman Sea belong to *L. olivacea*, with unusual lateral scute patterns, rather than *L. kempii*.

The lateral scute pattern of *L. olivacea* is much more diverse when compared to *L. kempii*, i.e., *L. olivacea* has both symmetric and asymmetric patterns that could vary from 3/5 to 9/9, but the main pattern may have been different for each location [14,39]. As such, in the Atlantic Ocean, the main pattern in Surinam was 7/7, while it was 6/6 for the Honduras population [39]. For the Sri Lankan or Ceylon populations in the Indian Ocean, Pritchard [39] found that from a total of 378 individuals, 94 individuals exhibited 6/6 as the main pattern, while 7/7 was the second pattern for the other individuals. In addition, a recent study conducted by Cherepanov and Malashichev [14] found that the highest proportion of newborn *L. olivacea* in Sri Lanka presented both 6/6 and 7/7 as the main patterns, accounting for 42.2%. Symmetrical patterns 5/5 (12.1%), asymmetrical patterns 5/6 (2.3%), 5/7 (0.15%), and 6/5 (1.5%) had also been observed in smaller numbers. Because there were such small numbers of these within the population, it is uncommon and scarce to see an individual at the adult stage of growth with rare carapace phenotypes. 

Conversely, the constancy of the five-paired lateral scute pattern was well known for *L. kempii* along the west coast of Florida and along the Rancho Nuevo beach in the Gulf of Mexico, which accounted for more than 95% of the turtles that displayed symmetrical five-paired patterns [40,41]. The occurrence of other patterns, including symmetrical 6/6 as well as asymmetrical 5/6 and 6/5, which were similar to *L. olivacea*, were also recorded for some individuals, but only in small numbers [40]. Thus, because the carapace pattern is shared amongst the *Lepidochelys* species, there were several cases of suspicious identification when only external morphology was used; this was particularly true in the overlapping areas. For example, at the Strait of Gibraltar in the Atlantic Ocean, Rojo–Nieto, et al. [42] reported a stranding event involving a *Lepidochelys*-like sea turtle with a 6/6 lateral scute pattern that was identified as *L. olivacea* based on the carapace pattern; however, this was later identified as *L. kempii* using the molecular markers of the mtDNA control region [43]. For the unusual distribution range of *L. kempii*, there was one report of live-stranded *Lepidochelys*-like sea turtles with a lateral scute pattern of 5/5 in the Pacific Ocean of Indonesia, which were identified as *L. kempii* [44]. However, the southernmost distribution range of *L. kempii* was known to be limited to Martinique Island in the eastern Caribbean of the Atlantic Ocean [12,45]. In fact, those identifications were based only on external morphology without other supporting genetic information. Thus, it still cannot be concluded that *L. kempii* occurred outside of the Atlantic Ocean, especially to the great distance in the Indo-Pacific region, until this can be carefully reinvestigated using molecular genetic markers.

Accordingly, mtDNA is usually used as an alternative method for species identification when the condition of external morphology is difficult to observe [46]. In our study, the use of the 457 bp mtDNA control region was also sufficient to identify the species of the two *Lepidochelys*-like turtles. A study conducted by Bowen, et al. [47] determined that the 470 bp sequence of the mtDNA control region could be used to observe the biogeographic distribution model between *L. olivacea* and *L. kempii* in the world’s oceans. The outcomes of this study suggest that there was a strong geographic partitioning of the mtDNA lineages within the genus and that the Indo-West Pacific region is the source of the most recent radiation of *L. olivacea* rather than the East Pacific region, as they represent the central haplogroup in the parsimony network. In the recent study conducted by Vilaça, et al. [48], a total of 62 haplotypes were revealed to belong to the *Lepidochelys* species on a global scale. Among this, 53 haplotypes were observed for *L. olivacea*, while 9 haplotypes were found for *L. kempii* without shared haplotypes occurring between both species [48], which was similar to our phylogenetic tree results as each species formed their own monophyletic clade. Moreover, the haplogroups between two species are known to be highly divergent from each other as they are separated by 30 mutation steps [48]. Thus, these findings also support the BLAST results of our two sequence samples that highly resemble the sequences from the Indo-Pacific Ocean population, and they are unlikely to originate from outside of this area. 

Other than the mtDNA control region, the sequence of 547 bp RAG2 nDNA also provided informative genetic data that could be used to discriminate closely related *Lepidochelys* species from each other by one substitution site. The other nDNA markers, such as the brain-derived neurotrophic factor (BDNF), the oocyte maturation factor (Mos—Cmos), and recombination activating Gene 1 (RAG1), were also used to investigate the presence of interspecific variations in sea turtles in areas that are known to have high hybridization rates [18,20]. These markers were shown to be species-specific and could be used to effectively differentiate between species and hybrids; however, from our preliminary study, between *L. olivacea* and *L. kempii*, RAG2 indicated that they possessed a different haplotype from each other. In fact, the sequences of the genes for *L. kempii* are currently considered rare in the NCBI database, suggesting that re-evaluation of the haplotype should be performed when there are more available sequences of *L. kempii*.

In accordance with climate change, the expansion of new nesting sites for sea turtle species has been observed in some areas. This is evidenced by the recent occurrence of *L. olivacea*, *C. mydas*, and *E. imbricata* that nest in the islands of the Cabo Verde Archipelago, East Atlantic Ocean [49]. Accordingly, this could lead to an increase in gene flow among populations and an increase in the chance of incidences of hybridization, as have been observed for *L. olivacea* and *C. caretta* in Sergipe, Brazil [50]. In that study, the mtDNA haplotype of *L. olivacea* was identified in the nesting female *C. caretta*, which could have occurred as a result of an overlap in reproduction periods and distribution areas [50]. Similarly, perhaps, there has also been expansion of new *L. olivacea* populations, or other species with rare phenotypes, from outside areas into the Thai Andaman Sea, and thus, some gene flow could have occurred among populations or interspecific levels that may have led to an increased chance for the mixing of phenotypes and hybridization. 

In future studies, we suggest that other molecular tools should also be used to improve the resolution of genetic information. This would contribute to more precise species identification and provide a complete record of morphological characteristics for *L. olivacea* and other species that occur in Thai seas. This would involve using multiple loci of microsatellite DNA markers with larger sample sizes to investigate genetic variations or instances of hybridization within a given population [48]. In addition, identification of the whole genome would also be useful, as it would become a more cost-effective and efficient method for the genetic study of sea turtles [27,51]. This could provide greater genetic resolution than the partial sequences that have already been recorded for *L. kempii* [27]. Currently, studies focusing on the genetic information of *L. olivacea* in and around the Thai Andaman Sea have been rare. Thus, the monitoring of these populations with the unusual carapace phenotype would still be needed. This monitoring should be performed to identify the characteristics of a population, the population structure, and any potential occurrences of hybridization that occur between species within the Cheloniidae family.

## 5. Conclusions

In this study, the authors have been provided with a rare opportunity to measure the morphometrics as well as sample the mtDNA of the olive ridley turtles (*L. olivacea*) that display a previously undocumented scute pattern. This was a confirmation of the species of two *Lepidochelys*-like sea turtles that possessed unusual lateral scute patterns that were found in the Thai Andaman Sea. The mtDNA control region and nDNA RAG2 were successfully used to confirm the species of both as *L. olivacea.* The findings of this study have also raised awareness for the exploration of rare phenotypes in *L. olivacea* populations that reside in Thai Seas and can be used to support future studies focusing on the occurrence of the unusual lateral scute patterns of *L. olivacea* in the Indo-Pacific region.

## Figures and Tables

**Figure 1 biology-13-00500-f001:**
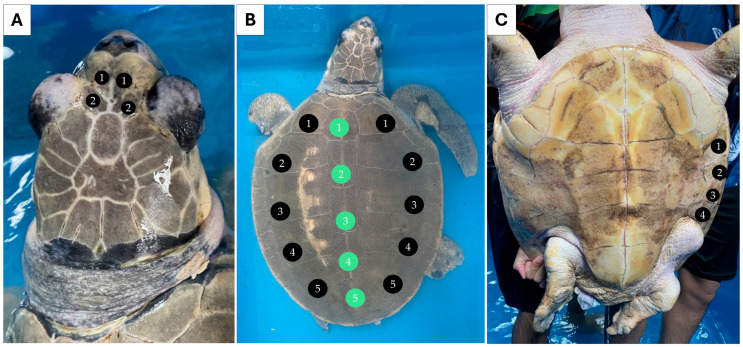
External morphology of individual 1 (sample GD) presenting symmetrical five paired lateral (5/5) and five vertebral scutes. Dorsal view of the head (**A**), dorsal view of the carapace. Black circles indicate the lateral scutes, while green circles indicate the vertebral scutes (**B**), and ventral view of the plastron (**C**).

**Figure 2 biology-13-00500-f002:**
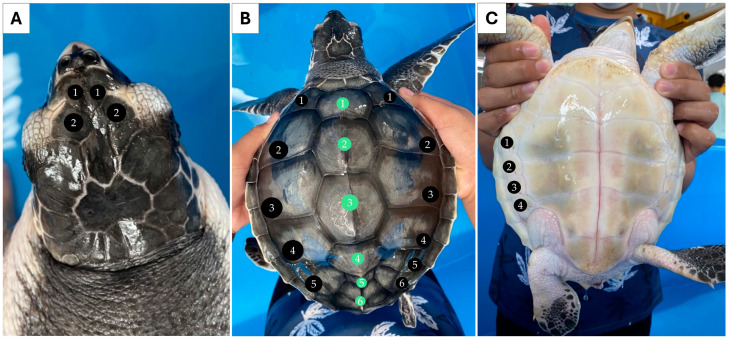
External morphology of individual 2 (sample HT) presenting asymmetrical lateral scute, five-left, six-right (5/6), and six vertebral scutes. Dorsal view of the head (**A**), dorsal view of the carapace. Black circles indicate the lateral scutes, while green circles indicate vertebral scutes (**B**), and ventral view of the plastron (**C**).

**Figure 3 biology-13-00500-f003:**
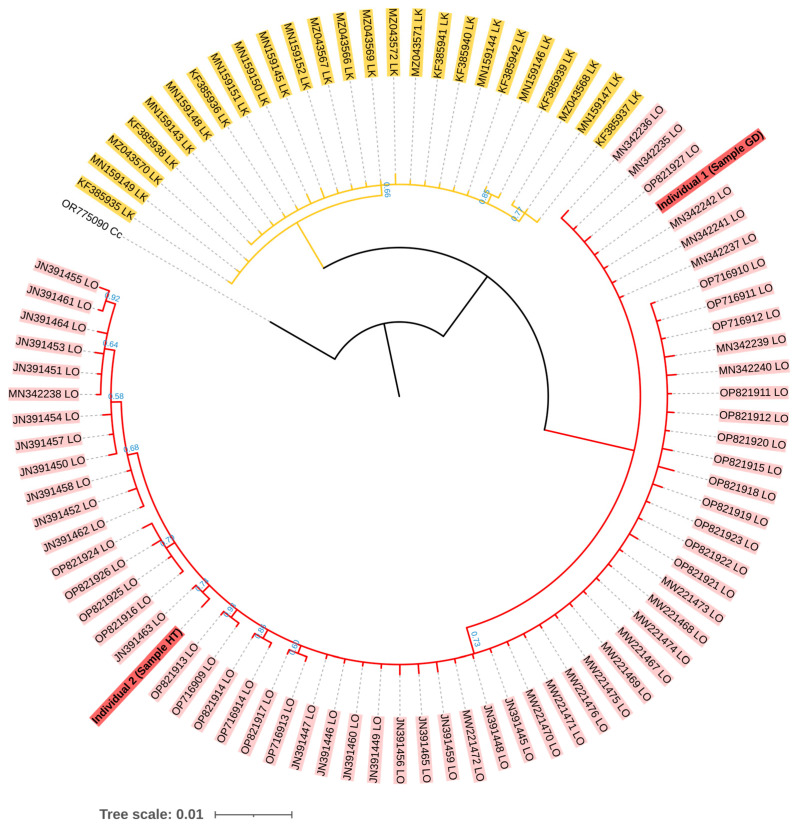
Bayesian phylogenetic tree of two individual turtles (accession numbers PP908976 and PP908977) based on the GTR evolutionary model. The posterior probability (PP) values are shown below each branch. All branches had a PP value greater than 0.5. The branch without numbering indicates a PP value greater than 0.99. The accession numbers of each sequence are labeled at the tips. Cc = *Caretta caretta*, LO = *Lepidochelys olivacea*, and LK = *Lepidochelys kempii*.

**Table 1 biology-13-00500-t001:** Details of variable sites on 547 bp of consensus alignment of RAG2 found between *L. olivacea* and *L. kempii* when compared with the samples in this study.

Accession Number (Species)	Variable Site	Location	Reference
	180		
FJ039983 (*L. olivacea*)	C	Pacific Ocean	[18]
FJ039990 (*L. olivacea*)	C	Atlantic Ocean	[18]
FJ039997 (*L. kempii*)	T	Atlantic Ocean	[18]
PP916054 (Individual 1)	C	Thai Andaman Sea	This study
PP916055 (Individual 2)	C	Thai Andaman Sea	This study

## Data Availability

The accession numbers used in this study are shown in Figure 3, Table 1, Appendix A.

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
