# Peer review of "Unusual Patterns of Lateral Scutes in Two Olive Ridley Turtles and Their Genetic Assignment to the Thai Andaman Sea Populations of Lepidochelys olivacea Eschscholtz, 1829"

_biology, 2024, doi:10.3390/biology13070500_

Round 1

Reviewer 1 Report

Comments and Suggestions for Authors

Lines 42-46 is there value in including the families of these turtles?

Lines 56-57 Should the authors be listed as Almeida et al.?

Line 60 awkward wording with “occur in some genus”. Change to “occur in some genera” for the plural form

Lines 66-68 authors state that the appearance clearly differentiates this species from others only to contradict that in the next sentence

Lines 85-86 Is there no nesting data more recent? The values stated are from 1990, which is 34 years ago

Lines 63-95 am these paragraphs need to be reworked because the info doesn’t follow the topic sentences. For example, line 63 starts with L. olive ea being the smallest and most abundant turtle in the world’s oceans. But the paragraph goes on to detail which oceans another species inhabitants as well as the scute patten and its prevalence

Lines 96-105 this paragraph fits better in the discussion. The authors could phrase this paragraph as “the authors were provided a rare opportunity to measure morphometrics as well as sample mtDNA of turtles with a previously undocumented scute pattern

Lines 228-229 For the Pritchard reference, a total of 378 individuals were examined, but the N for the two scute patterns listed only add up to 159

Reviewer 2 Report

Comments and Suggestions for Authors

Dear Authors,

In my opinion, you done an excellent work writing manuscript, and defending your obtained results/findings. Indeed, I am convinced that despite low number of samples your obtained results are enough for the task that had to be carried out, and current research have value at least in the future investigations into marine turtles morphology, genetics, conservation, and species identification. Currently your manuscript still requires some more work but I think that after the minor revision your improved manuscript version should be accepted for publishing. Below I present to you my constructive comments and suggestions.

Title, Simple Summary and Abstract

I think that this title could stay as it is. In general, Simple Summary and Abstract are almost ok. My advice:

Line 13: write that using both by morphology and molecular tools.

Lines 24-27: (one sentence) should be transferred to Simple Summary, as it would explain used molecular tools to readers unfamiliar with such techniques, and balance Abstract.

Line 29: give highest percentage values for L. kempii.

Introduction, Material and methods, and Results

Line 48: body shape include head shape, carapace form and so on, so I advise to rewrite: …and color. For instance, head shape…

Lines 102-105: could stay here but it sounds more like discussion and/or conclusion.

Line 111: so in 2021 or in 2022, as written in the Abstract?

Lines 129-131: it is not clear whether it was Taq DNA Polymerase ready to use mix or not. In case it was not, where MgCl2? By the way, it would be better if you would provide not only concentrations but also ul of each part of 25 ul.

Lines 134-135: using F, R or both primers? No PCR products preparation/purification step before sending for DNA sequencing?

Discussion and Conclusions

Write last new discussion paragraph that consist information about the future research main perspectives, goals/aims/tasks. For instance, write about the necessity to use microsatellite DNA markers with larger sampling to research hybridization (current and historical) nuances among L. olivacea and L. kempii specimens. Also there should be information about the rationale to conduct whole genome research using common and unique individuals/phenotypes from L. olivacea and L. kempii species in near future.

References and Supplementary Material

Revise everything according journal requirements in the Reference List. In the Supplementary Material I see strange text differences but other than that it seems that everything is ok.
